# What makes instance discrimination good for transfer learning?

**Nanxuan Zhao**[1*]  **Zhirong Wu**[2*]  **Rynson W.H. Lau**[1]  **Stephen Lin**[2]
[1]City University of Hong Kong  [2]Microsoft Research Asia

http://nxzhao.com/projects/good_transfer/

## Abstract

Contrastive visual pretraining based on the instance discrimination pretext task has made significant progress. Notably, recent work on unsupervised pretraining has shown to surpass the supervised counterpart for finetuning downstream applications such as object detection and segmentation. It comes as a surprise that image annotations would be better left unused for transfer learning. In this work, we investigate the following problems: What makes instance discrimination pretraining good for transfer learning? What knowledge is actually learned and transferred from these models? From this understanding of instance discrimination, how can we better exploit human annotation labels for pretraining? Our findings are threefold. First, what truly matters for the transfer is low-level and mid-level representations, not high-level representations. Second, the intra-category invariance enforced by the traditional supervised model weakens transferability by increasing task misalignment. Finally, supervised pretraining can be strengthened by following an exemplar-based approach without explicit constraints among the instances within the same category.

## 1 Introduction

Recently, a remarkable transfer learning result with unsupervised pretraining was reported on visual recognition. The pretraining method MoCo (He et al., 2020) established a milestone by outperforming the supervised counterpart, with an AP of $46.6$ compared to $42.4$ on PASCAL VOC object detection. Supervised pretraining has been the de facto standard for finetuning downstream applications, and it is surprising that labels of one million images, which took years to collect (Deng et al., 2009), appear to be unhelpful and perhaps even harmful for transfer learning. This raises the question of why contrastive pretraining provides better transfer performance and supervised pretraining falls short.

The leading contrastive pretraining methods follow an instance discrimination pretext task (Dosovitskiy et al., 2015; Wu et al., 2018; He et al., 2020; Chen et al., 2020a), where the features of each instance are pulled away from those of all other instances in the training set. Invariances are encoded from low-level image transformations such as cropping, scaling and color jittering. With such low-level induced invariances (Wu et al., 2018; Chen et al., 2020a), strong generalization has been achieved to high-level visual concepts such as object categories on ImageNet. On the other hand, the widely adopted supervised pretraining method optimizes the cross-entropy loss over the predictions and the labels. As a result, training instances within the same category are drawn closer while the training instances of different categories are pulled apart.

Toward a deeper understanding of why contrastive pretraining by instance discrimination performs so well, we dissect the performance of both contrastive and supervised methods on a few downstream tasks. Our study begins by studying the effects of pretraining image augmentations, which are shown to be crucial for contrastive learning. We find that both contrastive and supervised pretraining benefit from image augmentations for transfer performance, while contrastive models rely on these low-level augmentations significantly. With proper augmentations, supervised pretraining may still prevail on the downstream task of object detection on COCO and semantic segmentation on Cityscapes.

---

*Contributed equally to this work.

We then examine the common belief that the high-level semantic information is the key to effective transfer learning (Girshick et al., 2014; Long et al., 2015). On unsupervised pretraining with different types of image sets, it is found that transfer performance is largely unaffected by the high-level semantic content of the pretraining data, whether it matches the semantics of the target data or not. Moreover, pretraining on synthetic data, whose low-level properties are inconsistent with real images, leads to a drop in transfer performance. These results indicate that it is primarily low-level and mid-level representations that are transferred. Additionally, we notice that unsupervised pretraining on a much smaller dataset only marginally degrades the transfer performance.

We also delve deeply to understand the large margin (with AP 48.5 over 46.2) on VOC object detection obtained by contrastive pretraining over supervised pretraining. First, detection errors of both methods are diagnosed using the detection toolbox (Hoiem et al., 2012). It is found that supervised pretraining is more susceptible than contrastive pretraining to localization error. Secondly, to understand the localization error, we examine how effectively images can be reconstructed from contrastive and supervised representations. The results show that supervised representations mainly model the discriminative parts of objects, in contrast to the more holistic modeling of contrastive representations pretrained to discriminate instances rather than classes. Both sets of experiments suggest that there exists a greater misalignment of supervised pretraining to the downstream tasks, which requires accurate localization and full delineation of the objects.

Based on these studies, we conclude that, in visual pretraining, not only it is less critical to transfer high-level semantic information, but learning to discriminate among classes might be misaligned with the downstream tasks. We thus hypothesize that the essential difference that makes supervised pretraining weaker (and instance discrimination stronger) is the common practice of minimizing intra-class variation. The crude assumption that all instances within one category should be alike in the feature space neglects the unique information from each instance that may have significance in downstream applications. To validate that overfitting semantics leads to weakened transferability, we explore a new supervised pretraining method that does not explicitly embed instances of the same class in close proximity of one another. Rather, we pull away the true negatives of each training instance without enforcing any constraint on the positives. This respects the data distribution in a manner that preserves the variations in the positives, and our new pretraining method is shown to yield consistent improvements for both ImageNet classification and downstream transfer tasks.

We expect these findings to have broad implications over a variety of transfer learning applications. As long as there exists any misalignment between the pretraining and downstream tasks (which is true for most transfer learning scenarios in computer vision), one should always be careful about overfitting to the supervised invariances defined by the pretraining labels. We further test on two other transfer learning scenarios: few-shot image recognition and facial landmark prediction. Both of them are found to align with the conclusions obtained from our previous study.

## 2   AN ANALYSIS FOR VISUAL TRANSFER LEARNING

We study the transfer performance of pretrained models for a set of downstream tasks: object detection on PASCAL VOC07, object detection and instance segmentation on MSCOCO, and semantic segmentation on Cityscapes. Given a pretrained network, we re-purpose the network architecture, and finetune all layers in the network with synchronized batch normalization. For object detection on PASCAL VOC07, we use the ResNet50-C4 architecture in the Faster R-CNN framework (Ren et al., 2015). Optimization takes 9k iterations on 8 GPUs with a batch size of 2 images per GPU. The learning rate is initialized to 0.02 and decayed to be 10 times smaller after 6k and 8k iterations. For object detection and instance segmentation on MSCOCO, we use the ResNet50-C4 architecture in the Mask R-CNN framework (He et al., 2017). Optimization takes 90k iterations on 8 GPUs with a batch size of 2 images per GPU. The learning rate is initialized to 0.02 and decayed to be 10 times smaller after 60k and 80k iterations as of the 1x optimization setting. For semantic segmentation on Cityscapes, we use the DeepLab-v3 architecture (Chen et al., 2017) with image crops of 512 by 1024. Optimization takes 40k iterations on 4 GPUs with a batch size of 2 images per GPU. The learning rate is initialized to 0.01 and decayed with a poly schedule. Detection performance is measured by averaged precision (AP) and semantic segmentation performance is measured by mean intersection over union (mIoU). Each pretrained model is also evaluated by ImageNet classification of linear readoff on the last layer features.

Table 1: The effects of pretraining image augmentations on the transfer performance for supervised and unsupervised models. The strongest result for each downstream task is marked bold.

| Pretraining | Pytorch Augmentation | ImageNet | VOC07 | COCO | | CityScapes |
|---|---|---|---|---|---|---|
| | | Acc | AP | $AP_{box}$ | $AP_{seg}$ | mIoU |
| Supervised | + RandomHorizontalFlip(0.5) | 70.9 | 43.4 | 38.6 | 33.7 | 78.0 |
| | + RandomResizedCrop(224) [1] | 77.5 | 45.5 | 38.9 | 33.9 | 78.7 |
| | + ColorJitter(0.4, 0.4, 0.4, 0.1) | 77.4 | 45.9 | **39.3** | **34.4** | 78.7 |
| | + RandomGrayscale(p=0.2) | **77.7** | 46.4 | 39.1 | 34.2 | 78.7 |
| | + GaussianBlur(0.1, 0.2) | 77.3 | 46.2 | 38.9 | 33.9 | **78.8** |
| Unsupervised | + RandomHorizontalFlip(0.5) | 6.4 | 32.3 | 34.2 | 30.6 | 72.7 |
| | + RandomResizedCrop(224) | 53.0 | 43.2 | 36.8 | 32.3 | 76.6 |
| | + ColorJitter(0.4, 0.4, 0.4, 0.1) | 62.7 | 45.7 | 37.5 | 33.0 | 77.7 |
| | + RandomGrayscale(p=0.2) | 66.4 | 47.7 | 38.6 | 33.8 | 78.4 |
| | + GaussianBlur(0.1, 0.2) | 67.5 | **48.5** | 38.7 | 34.0 | 78.6 |

## 2.1 Effects of Image Augmentations on Pretraining

Contrastive learning is shown to depend on intensive image augmentations (views) for ImageNet classification. However, the effects of such image augmentations have not been carefully investigated for the downstream transfer tasks. In this section, we provide a detailed analysis on the effects of image augmentations for contrastive models, and supervised models as well.

We use MoCo-v2 (Chen et al., 2020b) for contrastive pretraining and the traditional cross entropy loss for supervised pretraining with the ResNet50 architecture. Both types of pretraining are optimized for 200 epochs with a cosine learning rate decay schedule for fair comparisons. The results are summarized in Table 1. First, unsupervised contrastive models consistently benefit much more from image augmentations than supervised models for ImageNet classification and all the transfer tasks. Supervised models trained merely with horizontal flipping may perform well. Second, supervised pretraining is found to also benefit from image augmentations such as color jittering and random grayscaling, but it is negatively affected by Gaussian blurring to a small degree. Third, unsupervised models outperform the supervised counterparts on PASCAL VOC, but underperform the supervised models on COCO and Cityscapes when proper image augmentations are applied on supervised models. This suggests that object detection on COCO and semantic segmentation on Cityscapes may benefit more from high-level information than PASCAL VOC.

## 2.2 Effects of Dataset Semantics on Pretraining

The strong performance of self-supervised pretraining on the linear classification protocol for ImageNet (He et al., 2020; Chen et al., 2020a) shows that the features capture high-level semantic representations of object categories. In supervised pretraining, it is a common belief that this high-level representation (Girshick et al., 2014; Long et al., 2015) is what transfers from ImageNet to the downstream tasks. Here, we challenge this conclusion by studying the transfer of contrastive models pretrained on images with little or no semantic overlap with the target dataset. These image datasets include faces, scenes, and synthetic street-view images. We also investigate how the size of the pretraining dataset affects transfer performance.

Consistently with the prior section, we use MoCo-v2 for contrastive pretraining and dedicated training pipeline for the supervised models on various datasets. Please refer to the Table 2 captions for detailed information about the annotation labels in each dataset. For the smaller datasets, we increase the number of training epochs and maintain the effective number of optimization iterations. The results are summarized in Table 2. First, it can be seen that, except for the results on the synthetic dataset Synthia, the transfer learning performance of contrastive pretraining is relatively unaffected by the pretraining image data, while supervised pretraining depends on the supervised semantics. Second, all the supervised networks are negatively impacted by the change of pretraining data except when the pretraining data has the same form of supervision as the target task, such as the Synthia and

---

[1] We optimally set the scale parameter to 0.08 for supervised pretraining and 0.2 for unsupervised pretraining.

Table 2: Transfer performance with pretraining on various datasets. "ImageNet-10%" denotes subsampling 1/10 of the images per class on the original ImageNet. "ImageNet-100" denotes subsampling 100 classes in the original ImageNet. Supervised pretraining uses the labels in the corresponding dataset, and unsupervised pretraining follows MoCo-v2. Supervised models for CelebA and Places are trained with identity and scene categorization supervision, while supervised models for COCO and Synthia are trained with semantic bounding box and segmentation supervision for detection and segmentation networks, respectively.

| Pretraining | Pretraining Data | #Imgs | Anno | ImageNet Acc | VOC07 AP | COCO AP$_{box}$ | AP$_{seg}$ | Cityscapes mIoU |
|---|---|---|---|---|---|---|---|---|
| | ImageNet | 1281K | object | 77.3 | 46.2 | 38.9 | 34.0 | 78.8 |
| | ImageNet-10% | 128K | object | 57.8 | 42.4 | 37.7 | 33.1 | 77.7 |
| | ImageNet-100 | 124K | object | 50.9 | 42.0 | 37.1 | 32.5 | 77.0 |
| Supervised | Places | 2449K | scene | 52.3 | 39.1 | 36.6 | 32.2 | 77.6 |
| | CelebA | 163K | identity | 30.3 | 37.5 | 36.4 | 32.2 | 76.5 |
| | COCO | 118K | bbox | 57.8 | 53.3 | 39.1 | 34.0 | 78.3 |
| | Synthia | 365K | segment | 30.2 | 40.2 | 37.3 | 32.9 | 76.5 |
| | ImageNet | 1281K | object | 67.5 | 48.5 | 38.7 | 34.0 | 78.6 |
| | ImageNet-10% | 128K | object | 58.9 | 45.5 | 38.6 | 33.9 | 78.1 |
| | ImageNet-100 | 124K | object | 56.5 | 45.6 | 38.3 | 33.6 | 77.8 |
| Unsupervised | Places | 2449K | scene | 57.1 | 46.7 | 38.4 | 33.6 | 78.8 |
| | CelebA | 163K | identity | 40.1 | 45.3 | 37.5 | 33.0 | 76.8 |
| | COCO | 118K | bbox | 50.6 | 46.1 | 38.4 | 33.7 | 78.3 |
| | Synthia | 365K | segment | 13.5 | 37.4 | 36.1 | 31.7 | 75.6 |

the Cityscapes datasets both sharing pixel-level annotations for semantic segmentation. Third, with the smaller amounts of pretraining data in ImageNet-10% and ImageNet-100, the advantage of unsupervised pretraining becomes more pronounced in relation to supervised models, which suggests stronger ability for generalization with less data for contrastive models.

Contrastive pretraining on faces and scenes achieves almost the same transfer results as pretraining on ImageNet. Since the face dataset has almost no semantic overlap with the VOC and COCO objects (besides the human category), transfer of high-level representations can be seen as extraneous. We further test unsupervised pretraining on the synthetic dataset Synthia (Ros et al., 2016), which exhibits low-level statistics different from real images. With this model, there is a substantial performance drop. We can therefore conclude that instance discrimination pretraining mainly transfers low-level and mid-level representations. In Table 2, we also test the linear readoff on ImageNet-1K classification using various pretrained models. These models perform very differently, suggesting that the last-layer features learned by contrastive training still overfit to the training data semantics.

## 2.3 Task Misalignment and Information Loss

A strong high-level representation is not critical for effective transfer, but this itself does not explain why contrastive pretraining yields better performance than supervised pretraining, specifically for object detection on PASCAL VOC. We notice that a larger performance gap exists on AP$_{75}$ than on AP$_{50}$ [2], which suggests that supervised pretraining is weaker at precise localization. For additional analysis, we use the detection toolbox (Hoiem et al., 2012) to diagnose detection errors. Figure 1 compares the error distributions of the transfer results on three example categories. We find that the detection errors of supervised pretraining models are more frequently the result of poor localization, where low IoU bounding boxes are mistakenly taken as true positives.

For further examination, we compare image reconstruction from the features of supervised and contrastive models. This reconstruction is performed by inverting the layer4 features (dimension of $7 \times 7 \times 2048$) using the deep image prior (Ulyanov et al., 2018). Specifically, given an image input $x_0$, we optimize a reconstruction network $r_\theta$ to produce a reconstruction $x$ that is close to the input

---

[2]please refer to the Appendix for detailed metrics on detection.

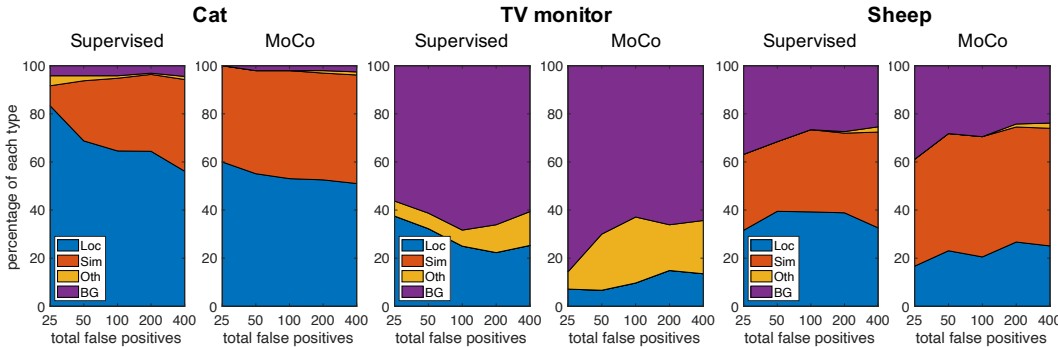

Figure 1: Analyzing detection error using the detection toolbox (Hoiem et al., 2012) on PASCAL VOC. Distribution of top-ranked false positive (FP) types for finetuning with supervised and unsupervised methods. Supervised pretraining models more frequently result in localization errors than unsupervised pretraining models. Each FP is categorized into 1 of 4 types: Loc—poor localization; Sim—confusion with a similar category; Oth—confusion with a dissimilar object category; BG—a FP that fires on background.

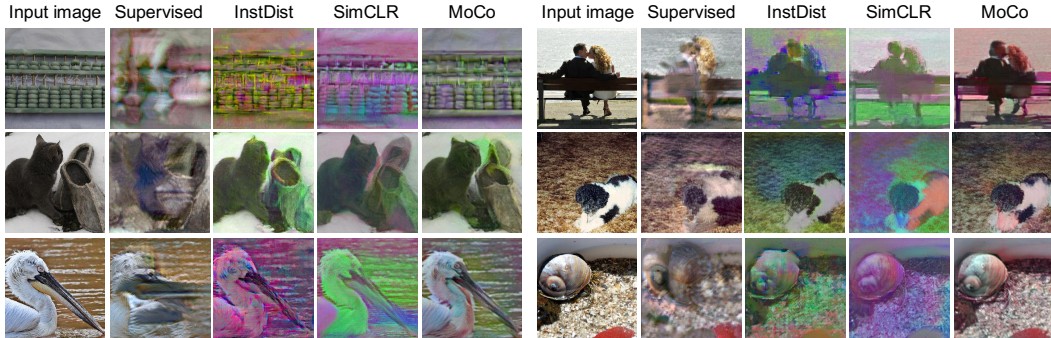

Figure 2: Image reconstruction by feature inversion. We use the method of deep image prior (Ulyanov et al., 2018) to reconstruct images by a pretrained network. Contrastive models allow for holistic reconstruction over the entire image, while supervised models lose information in many regions.

$x_0$ in the embedding space of a pretrained encoding network $f(\cdot)$,

$$\min_\theta E(f(x), f(x_0)), \quad x = r_\theta(z_0). \tag{1}$$

The input $z_0$ to the reconstruction network $r_\theta(\cdot)$ is fixed spatial noise, and the distance function $E(\cdot)$ is implemented as the $L2$ distance. The architecture of $r_\theta(\cdot)$ is an autoencoder network with six blocks for both the encoder and decoder, as detailed in the appendix. With this inversion method, we observe how well a pretrained network $g(\cdot)$ can recover image pixels from the features.

We visualize the reconstructions for both the supervised and contrastive pretrained networks. The investigated contrastive models include InstDisc, MoCo and SimCLR to show the generality of the results. In Figure 2, it is apparent that the contrastive network provides more complete reconstructions spatially, while the supervised network loses information over large regions in the images, likely because its features are mainly attuned to the most discriminative object parts, which are central to the classification task, rather than objects and images as a whole. The resulting loss of information may prevent the supervised network from detecting the full envelope of the object.

In Figure 2, we notice that for the contrastive models, the images are reconstructed at the correct scale and location. Though instance discrimination encodes invariances through spatial and scale transformations, features learned this way are still sensitive to these factors (Chen et al., 2020a). A possible explanation is that in order to make one instance unique from all other instances, the network strives to preserve as much information as possible. We also notice that the contrastive models find it difficult to reconstruct the hue color accurately. This is likely due to the broad space of colors and the intensive augmentations during pretraining.

Table 3: Exemplar-based supervised pretraining which does not enforce explicit constraints on the positives. It shows consistent improvements over the MoCo baselines by using labels.

| Methods | ImageNet Acc | VOC07 AP | COCO | | Cityscapes mIoU |
| | | | $AP_{box}$ | $AP_{seg}$ | |
| --- | --- | --- | --- | --- | --- |
| MoCo-v1 | 60.8 | 46.6 | 38.5 | 33.6 | 78.4 |
| Exemplar-v1 | 64.6 (+3.8) | 47.2 (+0.6) | 39.0 (+0.5) | 34.1 (+0.5) | 78.9 (+0.5) |
| MoCo-v2 | 67.5 | 48.5 | 38.7 | 34.0 | 78.6 |
| Exemplar-v2 | 68.9 (+1.4) | 48.8 (+0.3) | 39.4 (+0.7) | 34.4 (+0.4) | 78.8 (+0.2) |

## 3 A Better Supervised Pretraining Method

Annotating one million images in ImageNet provides rich semantic information which could be useful for downstream applications. However, traditional supervised learning minimizes intra-class variation by optimizing the cross-entropy loss between predictions and labels. By doing so, it focuses on the discriminative regions (Singh & Lee, 2017) within a category but at the cost of information loss in other regions. A better supervised pretraining method should instead pull away features of the true negatives for each instance without enforcing explicit constraints on the positives. This preserves the unique information of each positive instance while utilizing the label information in a weak manner.

We propose a new supervised pretraining method inspired by exemplar SVM (Malisiewicz et al., 2011), which trains an individual SVM classifier to separate each instance from its negatives. Unlike the original exemplar SVM which represents positives non-parametrically and negatives parametrically, our pretraining scheme models all instances in an non-parametric fashion in a spirit similar to instance discrimination (Wu et al., 2018). Concretely, we follow the framework of momentum contrast (He et al., 2020), where each training instance $x_i$ is augmented twice to form $x_i^q$ and $x_i^k$, which are fed into two encoders for embedding, $q_i = f_q(x_i^q), k_i = f_k(x_i^k)$. Please refer to MoCo (He et al., 2020) for details about the momentum encoders. But instead of discriminating from all other instances (Wu et al., 2018), the loss function uses the labels $y_i$ to filter the true negatives,

$$\mathcal{L}_{q_i} = -\log \frac{\exp(q_i^T k_i / \tau)}{\exp(q_i^T k_i / \tau) + \sum_{y_j \neq y_i} \exp(q_i^T k_j / \tau)}, \tag{2}$$

where $\tau$ is the temperature parameter. We set the baselines to be MoCo-v1 and MoCo-v2, and denote our corresponding methods as Exemplar-v1 and Exemplar-v2. Temperature $\tau = 0.07$ is used for Exemplar-v1, and $\tau = 0.1$ is used for Exemplar-v2 with ablations in the appendix. Experimental results are presented in Table 3. By filtering true negatives using semantic labels, our method consistently improves classification performance on ImageNet and transfer performance for all downstream tasks. This is in contrast to traditional supervised learning, where ImageNet performance is improved and its transfer performance is compromised. We note that the ImageNet classification performance of our exemplar-based training, $68.9\%$, is still far from the traditional supervised learning result of $77.3\%$. This leaves room for future research on even better classification and transfer learning performance.

## 4 Implications for Other Transfer Learning Scenarios

The presented studies focus on the transfer scenario of ImageNet pretraining. For other downstream applications, the nature of the task misalignment may differ. Thus, we additionally consider two other transfer learning scenarios to study the implications of overfitting to the supervised pretraining semantics and how it can be improved by our exemplar-based pretraining.

### 4.1 Few-shot Recognition

The first scenario is transfer learning for few-shot image recognition on the Mini-ImageNet dataset (Vinyals et al., 2016), where the pretext task is image recognition on the base 64 classes, and the downstream task is image recognition on five novel classes with few labeled images per class, either 1-shot or 5-shot. For the base classes, we split their data into training and validation sets to evaluate base task performance. The experimental setting largely follows a recent work (Chen et al.,

Table 4: 5-way few-shot recognition on Mini-ImageNet.

| Methods | Base classes Acc | Novel classes 1-shot | Novel classes 5-shot |
|---|---|---|---|
| Baseline | 82.3 | $51.75 \pm 0.80$ | $74.27 \pm 0.63$ |
| Supervised | 83.6 | $54.60 \pm 0.80$ | $74.50 \pm 0.65$ |
| MoCo-v2 | 75.3 | $52.14 \pm 0.73$ | $73.30 \pm 0.59$ |
| Exemplar-v2 | 79.9 | $55.33 \pm 0.75$ | $77.18 \pm 0.61$ |

Table 5: Facial landmark prediction on MAFL.

| Methods | Landmark error |
|---|---|
| Scratch | 24.6% |
| Supervised | 6.3% |
| MoCo-v2 | 5.8% |
| Exemplar-v2 | 5.8% |

Figure 3: Visual results of transfer learning for facial landmark prediction.

2019) for transfer learning. The pretrained network learned from the base classes is fixed, and a linear classifier is finetuned for 100 rounds on the output features for the novel classes.

As in our previous transfer learning study, we compare three pretraining methods: supervised cross entropy, unsupervised MoCo-v2 and supervised Exemplar-v2. Each method is trained with MoCo-v2 augmentations and optimized for 2000 epochs with a cosine learning rate decay scheduler for fair comparison. In finetuning the downstream task, since there exists much variance in the feature norms from different pretrained networks, we cross-validate the best learning rate for each method on the validation classes. Note that adding an additional batch normalization layer is problematic because as few as only 5 images (5-way 1-shot case) are available during finetuning.

We use the backbone network of ResNet18 (Chen et al., 2019) for the experiments. Results are shown in Table 4. Due to different optimizers and number of training epochs, our supervised pretraining protocol is stronger than the baseline protocol (Chen et al., 2019), leading to better results. The unsupervised pretraining method MoCo-v2 is weaker for both the base classes and the novel classes, suggesting that the pretraining task and the downstream task are well aligned semantically. Our exemplar-based approach obtains improvements over MoCo-v2 on the base classes, while outperforming the supervised baselines on the novel classes. This demonstrates that removing the explicit constraints on intra-class instances generalizes the model for better transfer learning on few-shot recognition.

## 4.2 FACIAL LANDMARK PREDICTION

We next consider the transfer learning scenario from face identification to facial landmark prediction on CelebA (Liu et al., 2018) and MAFL (Zhang et al., 2014). The pretext task is face identification on CelebA, and the downstream task is to predict five facial landmarks on the MAFL dataset. The facial landmark prediction is evaluated by the average euclidean distance of landmarks normalized by the inter-ocular distance.

As in the prior studies, we compare three pretraining methods: supervised cross entropy, unsupervised MoCo-v2 and supervised Exemplar-v2. Each method is trained with MoCo-v2 augmentations and optimized for 1400 epochs with a cosine learning rate decay scheduler. For landmark transfer, we finetune a two-layer network that maps the spatial output of ResNet50 features to landmark coordinates. The two-layer network contains a $1 \times 1$ convolutional layer that reduces 2048 channels to 128 and a fully connected layer, interleaved with LeakyReLU and batch normalization layers. We finetune all layers end-to-end for 200 epochs with a learning rate of $0.02$ and a batch size of 128.

The experimental results are summarized in Table 5. Unsupervised pretraining by MoCo-v2 outperforms the supervised counterpart for this transfer, suggesting that the task misalignment between face identification and landmark prediction is large. In other words, faces corresponding to the

same identity hardly reveal information about their poses. Our proposed exemplar-based pretraining approach weakens the influence of the pretraining semantics, leading to results that maintain the transfer performance of MoCo-v2. Qualitative results are displayed in Figure 3.

## 5 RELATED WORKS

Since the marriage of the ImageNet dataset (Deng et al., 2009) and deep neural networks (Krizhevsky et al., 2012), supervised ImageNet pretraining has proven to learn generic representations that facilitate a variety of applications such as high-level detection (Girshick et al., 2014; Sermanet et al., 2013) and segmentation (Long et al., 2015), low-level texture synthesis (Gatys et al., 2015), and style transfer (Gatys et al., 2016). ImageNet pretraining also works amazingly well under a large domain gap for medical imaging (Mormont et al., 2018) and depth estimation (Liu et al., 2015). The good transferability of ImageNet pretrained networks has been extensively studied (Agrawal et al., 2014; Azizpour et al., 2015). The transferability across each neural network layer has also been quantified for image classification (Yosinski et al., 2014), and a reduction of dataset size was found to have only a modest effect on transfer learning using AlexNet (Huh et al., 2016). In addition, a correlation between ImageNet classification accuracy and transfer performance has been reported (Kornblith et al., 2019), and the benefit of ImageNet pretraining has been shown to become marginal when the target task has a sufficient amount of data (He et al., 2019).

Beyond ImageNet transfer, there has been an effort to discover the structures and relations among tasks for general transfer learning (Silver & Bennett, 2008; Silver et al., 2013). Taskonomy (Zamir et al., 2018) builds a relation graph over 22 visual tasks and systematically studies the task similarities. In (Standley et al., 2019), task cooperation and task competition are quantitatively measured to improve transfer learning. Similar phenomena are observed that task misalignment may lead to negative transfer (Wang et al., 2019) and that the number of layers that tasks may share depends on the task similarity (Vandenhende et al., 2019). Other works (Dwivedi & Roig, 2019; Tran et al., 2019) explore alternative methods to measure the structure and similarity between tasks.

While most works study transfer learning based on supervised pretraining, our work focuses on analyzing transfer learning based on unsupervised pretraining, particularly on contrastive learning with the instance discrimination task (Dosovitskiy et al., 2015; Wu et al., 2018; He et al., 2020; Chen et al., 2020a). Over the years, the research community has achieved significant progress on self-supervised learning (Doersch et al., 2015; Doersch & Zisserman, 2017; Zhang et al., 2016; Gidaris et al., 2018; 2020) and contrastive learning (Oord et al., 2018; Zhuang et al., 2019; Tian et al., 2019; Hénaff et al., 2019; Zhao et al., 2020), closing the gap with supervised learning for ImageNet classification. More recent works (Goyal et al., 2019; Kolesnikov et al., 2019; Caron et al., 2019) make the attempts to scale unsupervised learning to uncurated data beyond ImageNet. Along with this, several research on contrastive learning (He et al., 2020; Misra & Maaten, 2020) report superior transfer results against the supervised counterparts on downstream tasks such as detection, segmentation and pose estimation. However, little is understood about why contrastive pretraining leads to improved transfer learning performance. Our work is the first to shed light on this, and it uses this understanding to elevate the performance of supervised pretraining.

## 6 CONCLUSION

This work provides an analysis for visual transfer learning to understand the recent advances of contrastive learning with the instance discrimination pretext task. Through this understanding, we also explore ways to better exploit the annotation labels for pretraining. Our main findings that help to understand the transfer are as follows:

- When finetuning the network over all layers end-to-end with thousands of data, it is mainly low- and mid-level representations that are transferred, not high-level representations. This suggests that contrastive representations learned on various datasets share a low- and mid-level representation which performs similarly and adapts quickly towards the target problem.

- The output features from contrastive models are not agnostic to the pretrained datasets, as they are still overfit to the high-level semantics of the dataset being trained on.

- Contrastive models learned by the instance discrimination pretext task contain rich information for reconstructing pixels from the output features. In order to discriminate among all instances, the network appears to learn a holistic encoding over the entire image.

- For supervised pretraining, the intra-class invariance encourages the network to focus on discriminative patterns and disregards patterns uninformative for classification. This may lose information which could be useful for the target task when there is task misalignment. An exemplar-SVM style pretraining scheme based on the instance discrimination framework is shown to improve generalization for downstream applications.

We expect these findings to have broad implications on other transfers. Our experiments on two other transfer scenarios confirm the generalization ability of the study. We hope that the presented studies provide insights that inspire better pretraining methods for transfer learning.

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

## A    EFFECTS OF PRETRAINING AND FINETUNING ITERATIONS

We also conduct experiments to examine the effects of pretraining optimization epochs and finetuning iterations. We show results in Figure 4, and find that longer optimization during pretraining consistently improves detection transfer for both supervised and unsupervised models. This suggests that overfitting is not an issue for either pretraining method. Unsupervised pretraining is seen to converge much faster during pretraining, and supervised pretrained models tend to converge faster in the initial iterations of detection finetuning but may not converge optimally.

We notice that supervised pretraining benefits from more optimization epochs. To explore the limit of supervised pretraining, we investigate larger numbers of supervised pretraining epochs. In Table 6, supervised pretraining continues to improve performance until 800 epochs, but may suffer from overfitting as indicated by the performance on ImageNet classification. For detection transfer, the improved supervised pretraining still falls short MoCo on AP and $AP_{75}$, while it outperforms MoCo on $AP_{50}$. This may possibly be due to the superior semantic classification ability of supervised models. Further discussion of the results are beyond the scope of the paper.

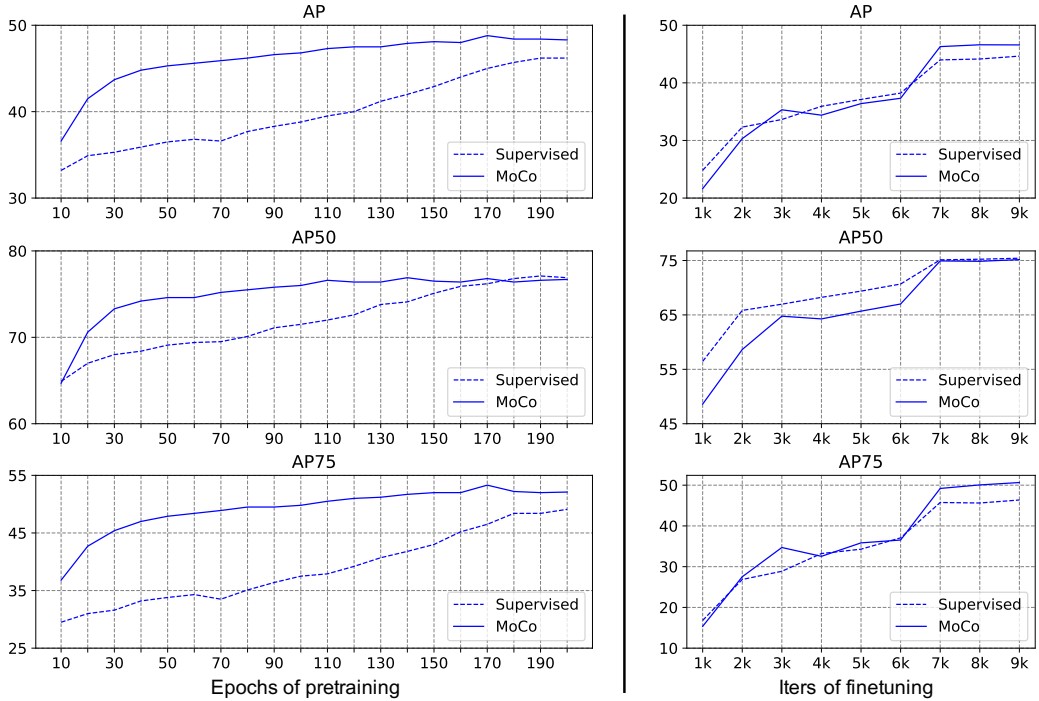

Figure 4: Performance at intermediate pretraining checkpoints and finetuning checkpoints.

Table 6: Longer supervised pretraining for object detection transfer on PASCAL VOC.

| Pretraining | ImageNet | VOC07 detection | | | VOC0712 detection | | |
|---|---|---|---|---|---|---|---|
| Epochs | Acc | AP | $AP_{50}$ | $AP_{75}$ | AP | $AP_{50}$ | $AP_{75}$ |
| 90 | 75.5 | 45.4 | 76.3 | 47.0 | 54.8 | 82.1 | 60.4 |
| 200 | 77.3 | 46.0 | 76.7 | 48.3 | 55.4 | 82.3 | 61.6 |
| 400 | 77.8 | 47.7 | 78.0 | 50.7 | 56.1 | 82.9 | 62.8 |
| 800 | 77.7 | 47.6 | 77.5 | 51.0 | 56.4 | 82.7 | 62.9 |
| MoCo | 67.5 | 48.5 | 76.8 | 52.7 | 56.9 | 82.2 | 63.5 |

## B    EFFECTS OF IMAGE AUGMENTATIONS ON PRETRAINING

We show full results of object detection on PASCAL VOC07, object detection and instance segmentation on MSCOCO, and semantic segmentation on Cityscapes in Table 7.

Table 7: The effects of pretraining image augmentations on the transfer performance for supervised and unsupervised models.

| Pytorch Augmentation | Supervised | | | | Unsupervised | | | |
|---|---|---|---|---|---|---|---|---|
| | ImageNet | VOC07 detection | | | ImageNet | VOC07 detection | | |
| | Acc | AP | $AP_{50}$ | $AP_{75}$ | Acc | AP | $AP_{50}$ | $AP_{75}$ |
| + RandomHorizontalFlip(0.5) | 70.9 | 43.4 | 74.0 | 44.5 | 6.4 | 32.3 | 58.3 | 31.4 |
| + RandomResizedCrop(224) | 77.5 | 45.5 | 76.2 | 47.4 | 53.0 | 43.2 | 71.2 | 45.4 |
| + ColorJitter(0.4, 0.4, 0.4, 0.1) | 77.4 | 45.9 | 76.7 | 48.0 | 62.7 | 45.7 | 74.4 | 48.6 |
| + RandomGrayscale(p=0.2) | 77.7 | 46.4 | 77.3 | 49.0 | 66.4 | 47.7 | 76.0 | 51.5 |
| + GaussianBlur(0.1, 0.2) | 77.3 | 46.2 | 76.8 | 48.9 | 67.5 | 48.5 | 76.8 | 52.7 |

| Supervised | | | | | | Unsupervised | | | | | |
|---|---|---|---|---|---|---|---|---|---|---|---|
| COCO detection | | | COCO segmentation | | | COCO detection | | | COCO segmentation | | |
| AP | $AP_{50}$ | $AP_{75}$ | AP | $AP_{50}$ | $AP_{75}$ | AP | $AP_{50}$ | $AP_{75}$ | AP | $AP_{50}$ | $AP_{75}$ |
| 38.6 | 58.5 | 41.7 | 33.7 | 55.1 | 35.9 | 34.2 | 52.7 | 36.7 | 30.6 | 49.9 | 32.4 |
| 38.9 | 59.3 | 41.6 | 34.0 | 55.7 | 36.0 | 36.8 | 56.1 | 39.7 | 32.3 | 52.9 | 34.4 |
| 39.3 | 59.6 | 42.3 | 34.4 | 56.1 | 36.4 | 37.5 | 56.9 | 40.5 | 33.0 | 54.0 | 35.0 |
| 39.1 | 59.2 | 42.0 | 34.2 | 55.6 | 36.4 | 38.6 | 58.0 | 41.9 | 33.8 | 54.8 | 36.0 |
| 38.9 | 59.1 | 41.8 | 33.9 | 55.4 | 35.9 | 38.7 | 58.1 | 42.0 | 34.0 | 55.1 | 36.4 |

| Supervised | | | Unsupervised | | |
|---|---|---|---|---|---|
| Cityscapes Segmentation | | | | | |
| mIoU | mAcc | aAcc | mIoU | mAcc | aAcc |
| 78.0 | 85.2 | 96.0 | 72.7 | 81.3 | 95.3 |
| 78.7 | 85.6 | 96.1 | 76.6 | 84.2 | 95.9 |
| 78.7 | 85.9 | 96.1 | 77.7 | 85.2 | 96.0 |
| 78.7 | 85.6 | 96.1 | 78.4 | 85.7 | 96.1 |
| 78.8 | 85.8 | 96.1 | 78.6 | 85.7 | 96.2 |

## C  EFFECTS OF DATASET SEMANTICS ON PRETRAINING

We report full transfer performance with pretraining on various datasets in Table 8. We also provide a visualization of various datasets for training these models in Figure 5.

## D  DETAILS ON IMAGE RECONSTRUCTION BY INVERTING FEATURES

### D.1  METHOD DETAILS

We use the same architecture for the reconstruction network $r_\theta(\cdot)$ as in the original deep image prior paper. It is an encoder-decoder network with the following architecture. Let $C_k^m$ denote a Convolution-BatchNorm-LeakyReLU layer with $k$ channels and $m \times m$ spatial filters; $CD_k^m$ denote a Convolution-Downsample-BatchNorm-LeakyReLU layer, and $CU_k^m$ denote a Convolution-BatchNorm-LeakyReLU-Upsample layer. We use a stride of 2 for both the upsampling and downsampling layers.

**Encoder:** $CD_{16}^7 - C_{16}^7 - CD_{32}^7 - C_{32}^7 - CD_{64}^5 - C_{64}^5 - CD_{128}^5 - C_{128}^5 - CD_{128}^3 - C_{128}^3 - CD_{128}^3 - C_{128}^3$

**Decoder:** $C_{16}^7 - CU_{16}^7 - C_{32}^7 - CU_{32}^7 - C_{64}^5 - CU_{64}^5 - C_{128}^5 - CU_{128}^5 - C_{128}^3 - CU_{128}^3 - C_{128}^3 - CU_{128}^3$

The input $z_0 \in R^{H \times W \times 32}$ is initialized with uniform noise between 0 and 0.1. For each image, the optimization takes 3000 iterations of an Adam optimizer with a learning rate of 0.001.

### D.2  EVALUATING RECONSTRUCTIONS BY PERCEPTUAL METRICS

To measure the reconstruction quality quantitatively, we calculate the perceptual distance between the reconstruction and the input image, using a deep learning based approach (Zhang et al. (2018)) with a SqueezeNet network. We randomly select one image per class from the ImageNet validation

Table 8: Transfer performance with pretraining on various datasets. "ImageNet-10%" denotes subsampling 1/10 of the images per class on the original ImageNet. "ImageNet-100" denotes subsampling 100 classes in the original ImageNet. Supervised pretraining uses the labels in the corresponding dataset, and unsupervised pretraining follows MoCo-v2. Supervised models for CelebA and Places are trained with identity and scene categorization supervision, while supervised models for COCO and Synthia are trained with semantic bounding box and segmentation supervision for detection and segmentation networks, respectively.

| Pretraining Data | #Imgs | Annotation | Supervised | | | | Unsupervised | | | |
|---|---|---|---|---|---|---|---|---|---|---|
| | | | ImageNet | VOC07 detection | | | ImageNet | VOC07 detection | | |
| | | | Acc | AP | $AP_{50}$ | $AP_{75}$ | Acc | AP | $AP_{50}$ | $AP_{75}$ |
| ImageNet | 1281K | object | 77.3 | 46.2 | 76.8 | 48.9 | 67.5 | 48.5 | 76.8 | 52.7 |
| ImageNet-10% | 128K | object | 57.8 | 42.4 | 73.5 | 43.1 | 58.9 | 45.5 | 74.4 | 48.0 |
| ImageNet-100 | 124K | object | 50.9 | 42.0 | 72.4 | 43.3 | 56.5 | 45.6 | 73.9 | 48.5 |
| Places | 2449K | scene | 52.3 | 39.1 | 70.0 | 38.7 | 57.1 | 46.7 | 74.9 | 50.2 |
| CelebA | 163K | identity | 30.3 | 37.5 | 66.1 | 36.9 | 40.1 | 45.3 | 72.4 | 48.4 |
| COCO | 118K | bbox | 57.8 | 53.3 | 80.3 | 59.5 | 50.6 | 46.1 | 74.5 | 49.4 |
| Synthia | 365K | segment | 30.2 | 40.2 | 70.3 | 40.2 | 13.5 | 37.4 | 65.0 | 37.2 |

| Supervised | | | | | | Unsupervised | | | | | |
|---|---|---|---|---|---|---|---|---|---|---|---|
| COCO detection | | | COCO segmentation | | | COCO detection | | | COCO segmentation | | |
| AP | $AP_{50}$ | $AP_{75}$ | AP | $AP_{50}$ | $AP_{75}$ | AP | $AP_{50}$ | $AP_{75}$ | AP | $AP_{50}$ | $AP_{75}$ |
| 38.9 | 59.1 | 41.8 | 33.9 | 55.4 | 35.9 | 38.7 | 58.1 | 42.0 | 34.0 | 55.1 | 36.4 |
| 37.7 | 57.5 | 40.5 | 33.1 | 54.3 | 35.1 | 38.6 | 58.0 | 41.7 | 33.9 | 54.9 | 36.0 |
| 37.1 | 56.6 | 40.1 | 32.5 | 53.3 | 34.5 | 38.3 | 57.7 | 41.6 | 33.6 | 54.5 | 35.5 |
| 36.6 | 56.3 | 39.1 | 32.2 | 53.1 | 34.1 | 38.4 | 58.0 | 41.3 | 33.6 | 54.5 | 35.7 |
| 36.4 | 55.5 | 39.4 | 32.2 | 52.2 | 34.5 | 37.5 | 56.5 | 40.3 | 33.0 | 53.5 | 35.3 |
| 39.1 | 58.9 | 42.3 | 34.0 | 55.5 | 36.2 | 38.4 | 58.0 | 41.6 | 33.7 | 54.6 | 35.8 |
| 37.3 | 57.1 | 40.4 | 32.9 | 53.8 | 35.0 | 36.1 | 55.0 | 38.6 | 31.7 | 51.9 | 33.7 |

| Supervised | | | Unsupervised | | |
|---|---|---|---|---|---|
| Cityscapes Segmentation | | | | | |
| mIoU | mAcc | aAcc | mIoU | mAcc | aAcc |
| 78.8 | 85.8 | 96.1 | 78.6 | 85.7 | 96.2 |
| 77.7 | 85.0 | 96.0 | 78.1 | 85.6 | 96.1 |
| 77.0 | 84.5 | 95.9 | 77.8 | 85.2 | 96.1 |
| 77.6 | 85.0 | 96.0 | 78.8 | 86.2 | 96.1 |
| 76.5 | 84.3 | 95.9 | 76.8 | 84.4 | 95.9 |
| 78.3 | 85.5 | 96.0 | 78.3 | 85.6 | 96.1 |
| 76.5 | 84.1 | 95.9 | 75.6 | 83.6 | 95.8 |

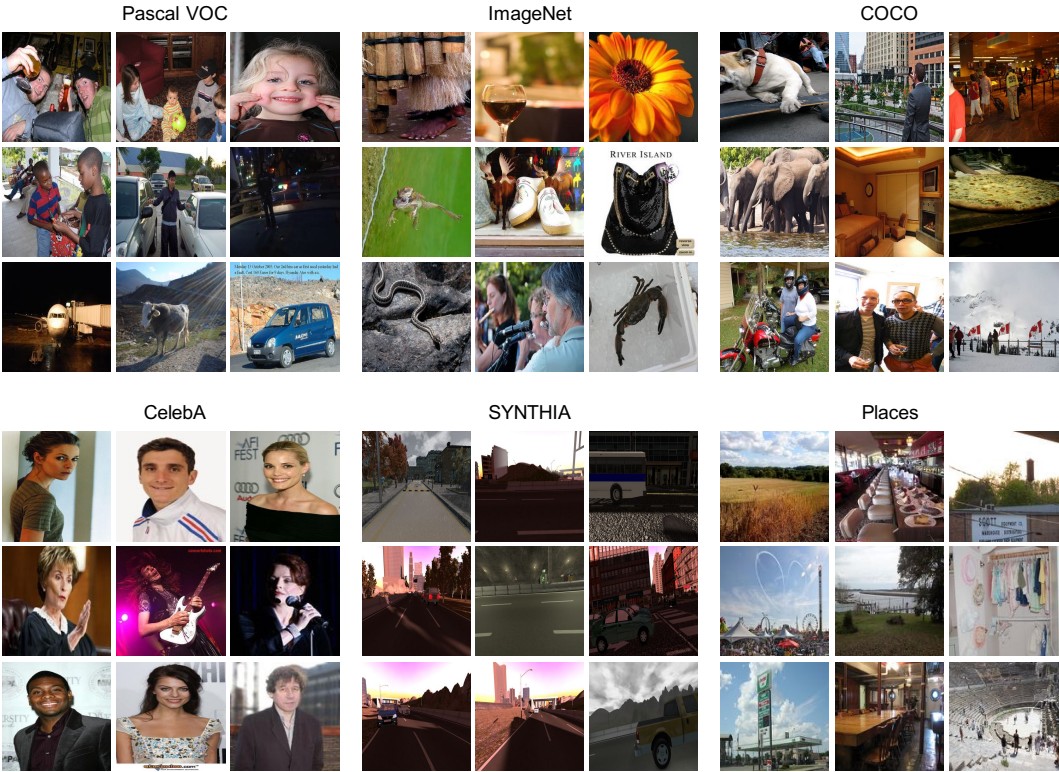

Figure 5: Example images of various datasets used for the pretraining study.

set for 1000 images in total. The average distance of reconstructions using MoCo is $5.59$, while it is $6.43$ for the supervised network. We provide a scatter plot of perceptual distance from individual reconstructions. In Figure 6, we can see that the reconstructions generated by MoCo are generally closer to the original images than those generated by the supervised method.

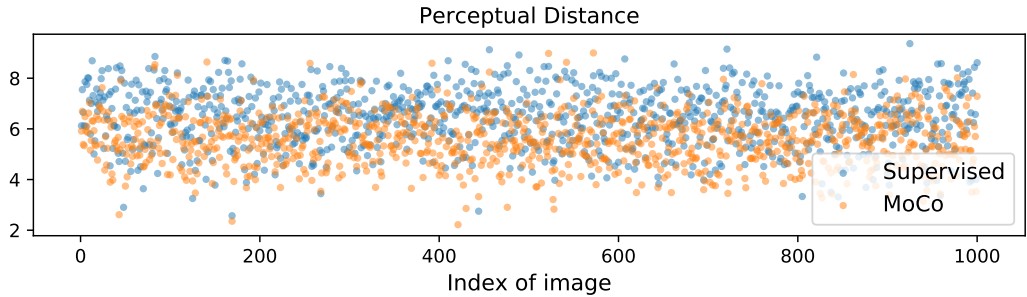

Figure 6: Perceptual distance between the reconstruction and the original image on 1000 validation images.

# E    MORE RESULTS ON EXEMPLAR-BASED SUPERVISED PRETRAINING

We show full transfer performance of our proposed Exemplar-based supervised pretraining in Table 9.

Since our Exemplar pretraining uses a different set of parameters from MoCo, we provide an ablation study over the parameter $k$ and $\tau$ for ImageNet linear readout in Table 10.

Table 9: Exemplar-based supervised pretraining which does not enforce explicit constraints on the positives. It shows consistent improvements over the MoCo baselines by using labels.

| Methods | ImageNet | VOC07 detection | | | Cityscapes segmentation | | |
|---|---|---|---|---|---|---|---|
| | Acc | AP | $AP_{50}$ | $AP_{75}$ | mIoU | mAcc | aAcc |
| MoCo-v1 | 60.8 | 46.6 | 74.9 | 50.1 | 78.4 | 85.6 | 96.1 |
| Exemplar-v1 | 64.6 | 47.2 | 76.0 | 50.6 | 78.9 | 86.0 | 96.2 |
| MoCo-v2 | 67.5 | 48.5 | 76.8 | 52.7 | 78.6 | 85.7 | 96.2 |
| Exemplar-v2 | 68.9 | 48.8 | 77.2 | 53.1 | 78.8 | 85.9 | 96.2 |

| Methods | COCO detection | | | COCO segmentation | | |
|---|---|---|---|---|---|---|
| | AP | $AP_{50}$ | $AP_{75}$ | AP | $AP_{50}$ | $AP_{75}$ |
| MoCo-v1 | 38.5 | 58.3 | 41.6 | 33.6 | 54.8 | 35.6 |
| Exemplar-v1 | 39.0 | 58.7 | 42.0 | 34.0 | 55.4 | 36.3 |
| MoCo-v2 | 38.7 | 58.1 | 42.0 | 34.0 | 55.1 | 36.4 |
| Exemplar-v2 | 39.4 | 59.1 | 42.7 | 34.4 | 55.9 | 36.5 |

Table 10: An ablation study of parameter $k$ and $\tau$ for MoCo and Exemplar pretraining.

| Methods | k | $\tau$ | ImageNet acc |
|---|---|---|---|
| MoCo-v1 | 65536 | 0.07 | 60.8 |
| MoCo-v1 | 1M | 0.07 | 60.9 |
| Exemplar-v1 | 1M | 0.07 | 64.6 |
| Exemplar-v1 | 1M | 0.1 | 63.9 |
| MoCo-v2 | 65536 | 0.2 | 67.5 |
| MoCo-v2 | 1M | 0.1 | 66.9 |
| MoCo-v2 | 1M | 0.2 | 67.8 |
| Exemplar-v2 | 1M | 0.07 | 68.1 |
| Exemplar-v2 | 1M | 0.1 | 68.9 |
| Exemplar-v2 | 1M | 0.2 | 67.9 |

## F    ADDITIONAL RESULTS OF DIAGNOSING DETECTION ERROR

We provide a full analysis over 20 object categories on the VOC07 test set. For each category, a pie chart is given to show the distribution of four kinds of errors in top-ranked false positives. For each category, the false positives are chosen to be within the top $N$ detections, where $N$ is chosen to be the number of ground truth objects in each category. The four types of false positives include: poor localization (Loc), confusion with similar objects (Sim), confusion with other VOC objects (Oth), or confusion with background or unlabeled objects (BG). In Figure 7, we compare the error distribution between the MoCo results and supervised results. It is apparent that detection results from the MoCo pretrained model exhibits a smaller proportion of localization errors.

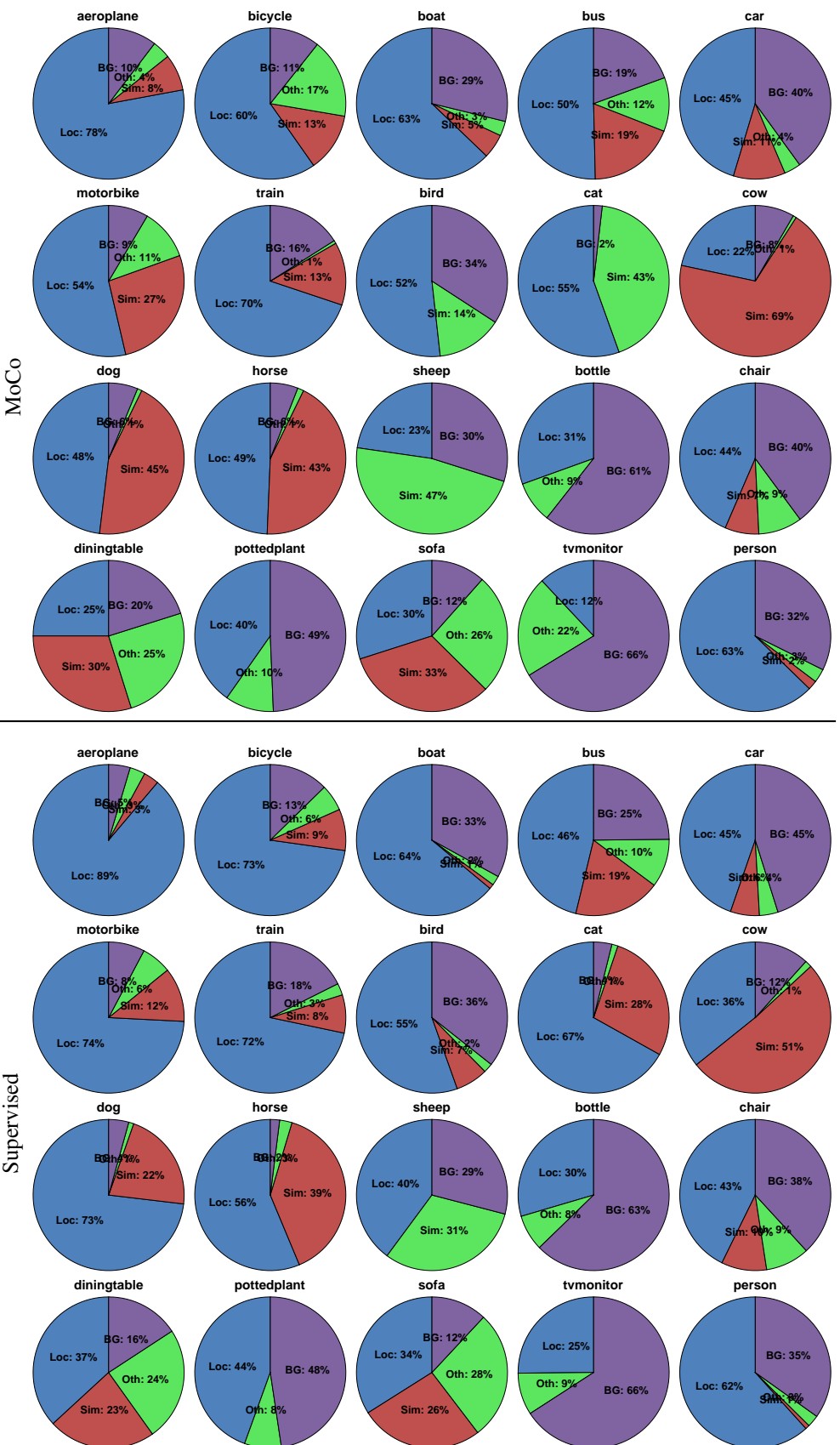

Figure 7: Distribution of four types of false positives for each category.

