# OpenReview forum: "What Makes Instance Discrimination Good for Transfer Learning?"
_ICLR.cc/2021/Conference — ICLR 2021 Poster_

### Official Review · AnonReviewer3 · 2020-10-27
**I think the paper addresses a relevant problem with strong experimental results. In my opinion, a very good paper.**

**Rating:** 8
**Confidence:** 4

**Review:**

Summary: The paper addresses the important topic of understanding why self-supervised learning methods show very good performance when used as pretraining for fine-tuning tasks. Authors analyse in detail the difference in performance between self-supervised and supervised pretraining and propose a new method to train model which improves over standard supervised models when used as pretraining.

Strengths:

- I think the paper addresses a relevant problem. Understanding difference between self-supervised and supervised pretraining is relevant to advance in this field. I particularly like the amount of evidence the paper provides to back all the claims, and the originality of some of the experiments such as Figure 2.

- The paper is well written and motivates the issue very well. I think it's particularly interesting to question traditional training techniques such as cross-entropy training, when the models are planed to be used for a different goal.

- The insights of the transferability experiments are useful for the community as point the strengths and weaknesses of each methods. I think it's good that authors analyse many different tasks such as classification, detection and segmentation with variate datasets.

- The proposed learning loss utilising the labels to produce the negatives is simple and seems to produce promising results according to Table 4 and Table 5.

- The extensive supplementary materials are also useful for additional information.

Weaknesses:

- It would have been interesting to see whether this presented results comparing MoCo with supervised pretraining hold with other self-supervised methods such as SimCLR. Do authors have any intuition on that? Are some of those effects from the particularities of the MoCo training or can we generalise to all self-supervised methods?

- Some Table references might be wrong. Section 2 refers to Table 7 and 8 which are in the supplemental (probably referring to Table 1 and Table 2).

- The face landmark task is a bit outside the main story of the paper. It is introduced very late in the paper and it is not clear where the proposed loss helps. I believe authors should clarify this points.


Conclusion: I believe the paper is strong enough for publication. I think it would be good for the reader if authors clarify a bit more the face landmark section and discuss a bit how would this compare to other self-supervison methods, but overall the paper is very good.

---

> ### Author Response · Authors · 2020-11-20
> **Responses to R3**
>
> We thank the reviewer for the valuable feedback.
>
>
> Q1: It would have been interesting to see whether this presented result comparing MoCo with supervised pretraining hold with other self-supervised methods such as SimCLR. Do authors have any intuition on that?
>
> In Figure 2, we examine three typical contrastive models (InstDisc, MoCo, SimCLR) on their ability to reconstruct the input image. The conclusion of preserving holistic information holds with these three models.
>
> For the rest of the transfer studies, though we are optimistic about generality, we are sorry that we cannot provide reliable experiments at this stage. For SimCLR, the original paper does not report the transfer protocol as well as the results for detection and segmentation. To the best of our efforts, we failed to obtain a competitive number using SimCLR compared with supervised pretrained models. Nevertheless, we agree with the reviewer on this promising direction to examine other self-supervised methods.
>
>
> Q2: Some Table references might be wrong. Section 2 refers to Table 7 and 8 which are in the supplemental (probably referring to Table 1 and Table 2).
>
> Thanks for pointing this out! These errors are fixed in the updated paper.
>
>
> Q3: The face landmark task is a bit outside the main story of the paper.
>
> Aside from detection transfer, face landmark prediction is another task where task misalignments occur between the pretraining and the downstream task. Traditional cross-entropy supervised pretraining may lose spatial information when pretrained to recognize faces, causing sub-optimal results for face landmark transfer. We introduce this in order to show that our proposed exemplar-based pretraining may mitigate the negative effect of supervised pretraining.

---

> > ### Comment · AnonReviewer3 · 2020-11-24
> > **Thanks!**
> >
> > Dear Authors,
> >
> > Thanks for your response to my concerns.
> >
> > I think all of them have been addressed and I will keep my original score.
> >
> > Thanks!

---

### Official Review · AnonReviewer4 · 2020-10-28

**Rating:** 5
**Confidence:** 4

**Review:**

This paper presents a detailed analysis of the task transfer abilities of an self-supervised representation, instance discrimination. The paper studies, in extensive detail, how these induced representations work for different tasks. The paper also proposes a new representation learning framework.

Overall, there are a lot of details in this paper, and it must have been a tremendous of work to put together. However, the "take home" message of the paper is not very clear. There are a lot of empirical findings throughout the paper, but it is left to the reader to decide what to do with the empirical findings. Which ones are important? What lessons should the field take away from these findings? This paper would be much stronger if it was re-organized to focus on the key take-away in this paper.

A central discussion point in the paper is about low, mid, or high-level features. These features were never fully defined. What is the difference between a mid or high-level feature?

---

> ### Author Response · Authors · 2020-11-20
> **Responese to R4**
>
> We thank the reviewer for the valuable feedback.
>
>
> Q1: The "take home" message of the paper is not very clear. This paper would be much stronger if it was reorganized to focus on the key take-away in this paper.
>
> The single focus of the paper is to answer the question – why contrastive learning transfers so well for downstream tasks. The empirical evidence from data augmentations, dataset semantics as well as task misalignments all shed light on this central problem. With insights and understandings from these studies, a straightforward implication for supervised pretraining is to use negatives without positives. The consistent improvements for the new supervised pretraining approach validate our prior findings about task misalignments for transfer learning. While the paper presents a spectrum of content, we believe all messages conveyed in the paper are tightly targeted to a central problem.
>
>
> Q2: What is the definition of low, mid, and high-level features?
>
> In this paper, we mainly distinguish high-level features from the others. Features are high-level if they pertain to the semantics of a particular dataset. In Table 2, since we manipulate the semantics of the datasets (e.g., faces, objects, scenes), we may control the high-level features that could be learned.  Features learned from solely faces cannot represent general high-level objects. In the paper, we do not further distinguish between low and mid-level features.

---

### Official Review · AnonReviewer2 · 2020-10-28
**What Makes Instance Discrimination Good for Transfer Learning?**

**Rating:** 7
**Confidence:** 4

**Review:**


**Summary:**

This work aims to explore why unsupervised contrastive pretraining works as well (if not better) than the tried-and-true Supervised ImageNet classification pretraining.  They explore a number of different transfer tasks to give some intuition:
1.  Interesting findings:
 - Augmentation doesn’t make much difference for supervised transfer (from imagenet) and is essential for unsupervised transfer, with the effect monotonically increasing as more augmentations are added.
 - Dataset semantics are less important for unsupervised pretraining:  They transfer from a variety of tasks, faces, objects, etc using supervised and unsupervised pretraining.
 - Imagenet pretraining leads to greater localization errors (using analysis of Hoiem 2012), and  more generally a loss of spatial information (tested via image reconstruction)
2. Propose a new supervised pretraining method to combine the idea of unsupervised contrastive training with supervised exemplar training.
 - Extend MoCo to use supervised labels so the loss doesn’t contrast examples from the same class.
 - This improves supervised transfer performance from ImageNet to the other tasks.
3. Finally, they look at the impact of this new pretraining on two other tasks
 - Few shot learning: The Supervised Exemplar model outperforms the unsupervised methods, and the original cross-entropy Imagenet supervised model.
 - Facial landmark prediction: Both the unsupervised and exemplar-supervised pretraining perform similarly and outperform either training from scratch or imagenet-supervised pretraining.  This again supports the observation that imagenet classification pretraining dilutes the spatial acuity of the model.

**Positives:**
- They introduce a supervised pretraining method that can transfer better than the unsupervised method and the original supervised imagenet method.
- Overall, this work is clearly written.
- Ultimately, I do believe I have a better understanding of the differences between the supervised and unsupervised pretrained models.

**Negatives:**
- The major insight into the differences is limited:  mainly that we pay a price when the low-level information is lost by the supervised pretrained model.

**Recommendation:**
These analysis papers are always tricky to rate -- often quite a bit of work goes into what seems like a small insight (maybe even obvious in retrospect).  However, I do think that this work is worthwhile for the community because 1) it shines light on a somewhat mysterious exciting new technique and 2) already shows how the findings are useful by using it to improve supervised pretraining, and a new vocabulary for evaluating pretraining techniques.

**Minor comments:**
Eq (2), should $v_i$ be $q_i$?

---

> ### Author Response · Authors · 2020-11-20
> **Responses to R2**
>
> We thank AnonReviewer2 for the positive assessment and detailed comments on our paper. We have fixed the typo of Eq (2) in the revised manuscript.

---

### Official Review · AnonReviewer1 · 2020-10-28
**An insightful paper**

**Rating:** 7
**Confidence:** 4

**Review:**

Summary:

The paper draws an interesting research question: why instance discrimination (ID) pretraining good for transfer learning?
The authors dissect ID for transfer learning by extensively comparing with supervised pretraining in several task combinations, so that they attempted to empirically answer what knowledge is learned by ID and transferred, what is differences with supervised pretraining, and when it is effective according to task relationships and dataset sizes.
Based on their findings, they propose a new supervised pretraining method, which has a good trade-off for transfer learning applications, and validates with other contexts such as few-shot classification and landmark localization.

The message has been deemed by researchers, but this paper shows empirical evidence with extensive experiments. Thus, the message contained in this work is worthwhile to report to our community.


Reasons for score:

Overall, I vote for accepting. I like the messages the authors want to convey through this work. If it presents and analyzes the effects according to task semantic relationship more specifically, it would have been a stronger paper. Also, it seems that there is room to improve the paper presentation further.


Pros:
- The paper opens and specifies which direction the transfer learning research should go.
- Also, the findings could be extended to few-shot, semi-supervised, and fully-supervised learning regimes.
- Extensive analysis and clear summarization of their findings

Cons:
- It would have been good if the presentation of the results is better organized and summarized. The current presentation is busy.


Other comments:
- This reviewer thinks that VOC and COCO datasets could be considered as a subset of the ImageNet dataset at a semantic level. Do the authors have any insight or conclusion from their experiments that can be drawn w.r.t the semantic inclusion relationship of the datasets?
- For the proposed supervised pretraining method, the authors intentionally draw a link from exemplar SVM. However, it turns out the final proposed method is a simple modification from [Wu et al. 2018]. It seems unnecessary to make a link with exemplar SVM.
- In sec 2.2, it seems that Table 8 is a typo, which is in the supplementary material.

---

> ### Author Response · Authors · 2020-11-20
> **Responses to R1**
>
> We thank the reviewer for the valuable feedback.
>
> Q1: Do the authors have any insight or conclusion from their experiments that can be drawn w.r.t the semantic inclusion relationship of the datasets?
>
> Supervised transfers rely heavily on the semantic inclusion/alignment of datasets, as they can suffer significantly from semantic misalignment of datasets. In contrast, unsupervised transfers are mostly agnostic to the semantics of the pretraining data. For example, on VOC object detection, supervised transfer drops sharply from 46.2 (ImageNet pretrain) to 39.1 (places pretrain), while unsupervised pretrain drops just marginally from 48.5 (ImageNet pretrain) to 46.7 (places pretrain). Similar observations can be found in COCO detection and Cityscapes segmentation.
> &nbsp;
>
> Q2: It would have been good if the presentation of the results is better organized and summarized. The current presentation is busy.
>
> In the revision, we revised the presentation of Table 1 and Table 2 to improve readability. We would be glad to incorporate any additional changes suggested by the reviewer.
> &nbsp;
>
> Q3: It seems unnecessary to make a link with exemplar SVM.
>
> We made this link because we were truly inspired by exemplar SVM paper to come up with the new pretraining approach. The idea to become free from explicit constraints on intra-class examples was pioneered by exemplar SVM. However, following the reviewer’s suggestion, we have removed the reference to exemplar SVM in the revision of the introduction section.
> &nbsp;
>
> Q4: In sec 2.2, it seems that Table 8 is a typo, which is in the supplementary material.
>
> Thanks for catching this! The paper has been updated to fix this problem.

---

### Comment · ~ZIYUN_LI1 · 2022-06-27
**Why not try supervised constrstive loss?**

Thanks for the interesting paper!
And I have a question about the supervised pertaining method you proposed.
In Section 3,  the loss function is a variant of constructive learning, and we can regard it combines supervised label knowledge and instance knowledge. And I am curious how about using supervised contrastive loss (Supervised Contrastive Learning), would it get similar results? Compared to contrastive loss, supervised contrastive loss focuses on true positive while your method focuses on true negative.
Why not consider both?

---

> ### Comment · ~Nanxuan_Zhao1 · 2022-07-01
> **Responses to the supervised contrastive loss**
>
> Thanks for your question. From the theoretical analyses and our empirical experiment, adding supervised contrastive loss on both true positive and true negative together is similar to the original supervised softmax loss.

---

> > ### Comment · ~ZIYUN_LI1 · 2022-07-04
> > **How about only considering true postive without true negative?**
> >
> > Thanks for answering.
> > Based on your empirical experiments, did you try to only use supervised contrastive loss without considering true negative?
> > Is the result similar to only focusing on true positive?

---

> > > ### Comment · ~Zhirong_Wu1 · 2022-07-04
> > > **True positives without negatives**
> > >
> > > Considering true positives without negatives will lead to collapse, all images falling into a single representation.

---

### Decision · Program_Chairs · 2021-01-07
**Final Decision**

**Decision:**

Accept (Poster)

**Comment:**

The paper aims at understanding why self-supervised/contrastive learning methods  transfer well when used as pretraining for fine-tuning downstream tasks  (compared to e.g., supervised pretraining based on the cross-entropy loss). Three reviewers recommend acceptance, whereas one reviewer recommends borderline rejection, arguing the take home message of the paper is not very clear. While this is a legitimate concern, the AC agrees with the majority that the paper does shed light on the differences between supervised and self-supervised pretraining (based on interesting empirical findings) and recommends acceptance.